# The Facile Synthesis and Application of Mesoporous Silica Nanoparticles with a Vinyl Functional Group for Plastic Recycling

**DOI:** 10.3390/ijms25042295

**Published:** 2024-02-15

**Authors:** Jong-tak Lee, Misun Kang, Jae Young Bae

**Affiliations:** Department of Chemistry, Keimyung University, Daegu 42601, Republic of Korea; atomos27@daum.net (J.-t.L.); misun.kang@gmail.com (M.K.)

**Keywords:** mesoporous silica nanoparticles, plastic recycling, vinyl functional groups, sodium silicate, vinyltrimethoxysilane

## Abstract

Due to growing concerns about environmental pollution from plastic waste, plastic recycling research is gaining momentum. Traditional methods, such as incorporating inorganic particles, increasing cross-linking density with peroxides, and blending with silicone monomers, often improve mechanical properties but reduce flexibility for specific performance requirements. This study focuses on synthesizing silica nanoparticles with vinyl functional groups and evaluating their mechanical performance when used in recycled plastics. Silica precursors, namely sodium silicate and vinyltrimethoxysilane (VTMS), combined with a surfactant, were employed to create pores, increasing silica’s surface area. The early-stage introduction of vinyl functional groups prevented the typical post-synthesis reduction in surface area. Porous silica was produced in varying quantities of VTMS, and the synthesized porous silica nanomaterials were incorporated into recycled polyethylene to induce cross-linking. Despite a decrease in surface area with increasing VTMS content, a significant surface area of 883 m^2^/g was achieved. In conclusion, porous silica with the right amount of vinyl content exhibited improved mechanical performance, including increased tensile strength, compared to conventional porous silica. This study shows that synthesized porous silica with integrated vinyl functional groups effectively enhances the performance of recycled plastics.

## 1. Introduction

The widespread use of plastic products worldwide has naturally led to the emergence of environmental pollution issues related to plastic waste [1,2,3]. Typically, after the use of plastic products, they are disposed of through landfilling or incineration, resulting in associated soil and air pollution, along with high disposal costs [4,5]. To mitigate these challenges, active research is being conducted on plastic waste recycling. However, currently, only about 9% of plastic waste produced globally is recycled, while the remaining 91% goes unrecycled [6]. Polyolefin, a thermoplastic resin, is commonly used as a raw material for plastics and is extensively employed in various facets of daily life [7,8,9,10]. Polyolefin-based polymer refers to chain or ring-like hydrocarbon polymers characterized by one or more unsaturated groups. Notable examples of polyolefins include polyethylene (PE), polypropylene (PP), polyisobutene, and polybutene. These versatile resins are widely employed in packaging materials, tableware, films, and numerous other applications [11,12,13,14]. Numerous studies have been conducted to facilitate the recycling of polyolefin-based thermoplastic materials. Among these, notable approaches include the introduction of inorganic nanoparticles [15,16,17,18,19], radical-induced cross-linking reactions [20,21], and the utilization of silane monomers for cross-linking [22,23]. Regarding the incorporation of inorganic nanoparticles, extensive research has been devoted to enhancing the mechanical properties of polyolefins by integrating nano-silica particles, with a particular emphasis on mesoporous silica [24,25,26]. It is generally observed that silica with a higher surface area leads to improved performance when introduced into waste plastics. However, commercially available silica often lacks a significantly high surface area, and the synthesis of mesoporous silica with a surface area exceeding 1000 m^2^/g can be challenging due to complex production methods and the associated high costs. Furthermore, there are limitations in achieving performance improvements solely through the physical dispersion of silica nanoparticles, necessitating further research in this domain and the introduction of peroxides into recycled plastics. Upon exposure to a specific temperature threshold, peroxides generate oxygen radicals that react with unreacted unsaturated groups, thereby increasing cross-link density. While this can enhance physical properties, the presence of residual unreacted peroxides may lead to adverse effects [27], making it a less desirable option. The use of silane monomers in cross-linking reactions involving unsaturated groups introduces silane monomers processing unsaturated groups into recycled plastics. These unsaturated groups of silanes react with the unsaturated group of the plastics, increasing the cross-linking density and receiving the inorganic properties of the silane. However, silane monomers are typically in liquid form, which can impose limitations during the mixing process. Moreover, relying solely on the introduction of silane monomers may have inherent constraints, necessitating further research efforts. In other words, there is a need for research aimed at increasing the concentration of inorganic components to enhance tensile strength while simultaneously addressing the compromise in flexibility performance resulting from the hindered cross-linking density due to the incorporation of inorganic additives.

In this research, we aim to enhance the physical performance of recycled plastic by synthesizing nano-sized silica materials and introducing vinyl functional groups into them to increase cross-linking density. Specifically, we refer to the synthesis method of mesoporous silica nanomaterials using sodium silicate solution that was previously studied, adapting it to leverage the excellent specific surface area of nano-sized silica [28,29]. Nevertheless, introducing the vinyl functional group poses a challenge in the conventional method, where vinyl silane is added after the synthesis of mesoporous silica nanomaterials, leading to potential pore blockage and a reduction in specific surface area [30]. Therefore, in this study, we aim to incorporate vinyl silane into the silica precursor in the initial stage of synthesizing mesoporous silica nanomaterials to allow the reaction to proceed. Research on the early-stage introduction of vinyl silane as a precursor using tetraethylorthosilicate has been widely conducted recently [31]. However, in our research, we distinguish ourselves by introducing sodium silicate solution as a precursor along with vinyl silane during synthesis. This approach represents a distinctive feature in the synthesis process and is a significantly cost-effective method for raw materials. Therefore, we aim to incorporate mesoporous silica nanomaterials with the introduced vinyl functional groups, synthesized using the aforementioned method, into recycled plastic to ensure the superior mechanical performance of the recycled plastic.

## 2. Results

Vinyl functional groups were introduced into porous silica by adjusting the volume ratio of sodium silicate to VTMS from 40:0 to 25:15, as shown in Table 1, and TEM images of the synthesized mesoporous silica are presented in Figure 1. When the vinyl group content exceeded 20 mL, gelation occurred and the reaction did not proceed. The TEM images confirm the successful formation of well-defined meso-sized pores in the vinyl-functionalized MS nanoparticles (V-MS). Based on the TEM images, it can be observed that the meso-sized pores within the particles are partially arranged in a hexagonal configuration. However, it becomes increasingly difficult to identify a regular pore arrangement as the vinyl silane content increases.

The low-angle XRD analysis data of the V-MS nanoparticles are shown in Figure 2. In the XRD result of V00-MS (the black line in Figure 2), which does not contain a vinyl functional group, a main peak corresponding to (100) is observed at around 2.06 in 2-theta. The small and broad peak approximately 3.91 in 2-theta, corresponding to (110), is also observed, confirming the hexagonal arrangement of the pore [32]. However, the incorporation of vinyl silane during sample preparation renders the (110) peak undetectable in the XRD data, clearly indicating a significant reduction in the hexagonal pore arrangement of the MS nanoparticles. Additionally, it is observed that the main peak associated with (100), indicated by the arrows in Figure 2, undergoes a noticeable rightward shift as the content of vinyl silane increases. It can be inferred that a reduction in the pore size of mesoporous silica nanoparticles takes place. The V15-MS sample, with the highest vinyl group content, shows broader peaks of XRD analysis compared to other samples, indicating an inconsistency in pore size. This can be attributed to the fact that the vinyl silane only possesses three alkoxysilane groups, leading to uneven silane polymerization when compared to other samples.

The N_2_ adsorption–desorption isotherm graphs for silica samples with incorporated vinyl functional groups are presented in Figure 3. It is evident that all four types of mesoporous silica nanoparticles exhibit a Type IV hysteresis loop pattern [33]. Furthermore, the presence of separated adsorption and desorption curves suggests the existence of pores within the silica. The detailed pore properties of the MS are summarized in Table 2.

As the amount of VTMS in the composite increases, it can be observed that the specific surface area decreases while the pore volume increases. Since VTMS possesses a three-functional alkoxy group in contrast to the sodium silicate, the cross-linking density between the silanes is lower than that of sodium silicate. This leads to the distribution of relatively large pores within the mesoporous silica nanoparticle. As a result, the increase in pore volume can be attributed to the decrease in specific surface area. While there does not seem to be a substantial difference in pore size, it can also be inferred that the presence of significant internal pores is suggested by the separation phenomenon between the adsorption and desorption curves, even at higher relative pressures (P/P_0_).

The FTIR analysis results of mesoporous silica nanoparticles incorporating vinyl groups (V-MS) are shown in Figure 4. Symmetric and asymmetric stretching vibrations of Si-O-Si are observed at 1057 cm^−1^ (the black arrow in Figure 4) and 1215 cm^−1^ (the blue arrow in Figure 4), respectively [31]. For the signal of vinyl group, a peak corresponding to the C=C double bond is detected at 1596 cm^−1^ (dashed line in Figure 4), and C-H peaks associated with the C=C double bond are observed at 3097 cm^−1^ and 3181 cm^−1^ (dotted line in Figure 4) [34]. In V00-MS, there is no observable peak associated with vinyl groups. It can be noted that as the VTMS content increases, the peaks related to vinyl groups also increase. Nevertheless, when comparing V10-MS and V15-MS, the peaks associated with vinyl groups are nearly identical, suggesting that there may be a limit to the extent of vinyl group incorporation. These results indicate that vinyl groups have been successfully introduced into mesoporous silica.

Based on the analysis results of the MS samples with vinyl groups, we proceeded to manufacture pellets by incorporating them into recycled PE, followed by characterizing their properties. To determine the optimal MS nanoparticles with vinyl group content, we conducted a series of tests using V05-MS samples at various concentrations, and the results are presented in Table 3.

The results show that with increasing V05-MS NP content, both tensile and flexural strengths improve compared to virgin recycled polyethylene (rePE). However, a decrease in elongation, elastic modulus, and flexural modulus is observed. The most suitable composition was found to be 1 part of V05-MS NPs per 100 parts of recycled PE, demonstrating the most superior characteristics. This can be explained by the fact that as the content of mesoporous silica with introduced vinyl functional groups increases, there are challenges with dispersion during the production of recycled PE, leading to a decrease in performance. This is because as the specific surface area of mesoporous silica increases, particularly due to its high surface energy, it induces aggregation [35]. A low melt index typically signifies a lower molecular weight of the polymer. As the silica content increases, the dispersibility diminishes, and undispersed silica particles interfere with the bonding between silica and polyethylene. This interference results in a reduction in melt index [36]. Therefore, while keeping the content of V-MS fixed at 1 part, experiments were conducted by varying the content of the vinyl functional group, and the results are presented in Table 4.

When various amounts of vinyl functional groups were introduced into mesoporous silica and mixed into recycled PE, V10-MS showed the best properties. As the vinyl content increases, there is a presumed enhancement in the physical properties of recycled PE, attributed to the escalating cross-linking density among vinyl functional groups. However, in the case of V15-MS, which introduced a similar amount of vinyl functional groups based on FTIR measurements, the physical properties improved compared to the pure recycled PE sample. In contrast, when compared to V10-MS, the physical properties actually decreased. This is attributed to the fact that, even with a similar amount content of the vinyl functional groups, the specific surface area of silica mesoporous particles, due to their inherent porosity, is reduced, resulting in a slight reduction in the physical properties of recycled PE. The introduction of inorganic silica nanoparticles typically results in an increase in tensile strength due to the inherent performance of these inorganic particles, even in the absence of direct bonding with recycled PE. This often leads to a reduction in the flexibility or properties requiring flexibility, as is commonly observed [37]. This can be observed in the performance indicators of recycled PE produced with VT00-MS, as evidenced in Table 4. However, in the case of introducing silica with incorporated vinyl functional groups into recycled PE, it can be observed that performance indicators requiring flexibility also increase. This phenomenon is attributed to the silica nanoparticles becoming homogeneously cross-linked through chemical bonding rather than physical dispersion, resulting in an increase in cross-linking density and a more homogeneous composite structure [38]. In conclusion, it can be stated that the introduction of mesoporous silica nanoparticles into recycled PE results in superior performance, with higher specific surface areas and higher vinyl group contents proving to be advantageous.

In Figure 5, the FTIR graph of recycled PE containing mesoporous silica is presented. The typical polyethylene FTIR peaks are similar, but the presence of synthesized mesoporous silica is confirmed by the Si-O-Si peak at 1100 cm^−1^ (the blue arrow in Figure 5). Notably, attention should be given to the stretching peak of C=C double bonds at 1640 cm^−1^ (the blue rectangular area in Figure 5). The peak for V05-MS_PE_A, which includes vinyl functional groups, significantly decreases compared to the peak for V00-MS_PE_A, which does not contain vinyl groups. This suggests that the functional groups of vinyl in mesoporous silica induce bonding between vinyl groups in recycled PE, leading to a decrease in the peak intensity.

Additionally, the more distinct Si-O-Si peak in V05-MS_PE_A compared to V00-MS_PE_A further supports this observation. Through the analysis of these results and the mechanical property values in Table 4, it can be concluded that the inclusion of vinyl functional groups in mesoporous silica contributes positively to the chemical bonding with recycled PE, thereby enhancing the overall properties of recycled PE.

## 3. Materials and Methods

### 3.1. Materials

In the synthesis of mesoporous silica incorporating vinyl functionalization, we employed materials procured from Kwangduk Industry Co. and Daejung Chemicals & Metals Co. (Siheung-si, Republic of Korea). Virgin polyethylene (KD-PE030) and recycled polyethylene grade A (KD-rePE_030A) were sourced from Kwangduk Industry Co. for use in cross-linking applications. For the synthesis of vinyl-functionalized mesoporous silica, sodium silicate (Na_2_SiO_3_, 20 wt.% of aqueous solution), hydrochloric acid (HCl, 35~37%), ethanol (CH_3_CH_2_OH), cetyltrimethyl ammonium chloride (CTACl, CH_3_(CH_2_)_15_N(Cl)(CH_3_)_3_, 25 wt.% of aqueous solution), and vinyltrimethoxysilane (VTMS, H_2_C=CHSi(OCH_3_)_3_, >98%) were purchased from Daejung Chemicals & Metals Co.

As initiators for cross-linking, dicumyl peroxide ([C_6_H_5_C(CH_3_)_2_]_2_O_2_, >99%) from Kwangduk Industry Co. was employed, and dibutyltin dilaurate ([CH_3_(CH_2_)_3_]Sn[O_2_C(CH_2_)_10_CH_3_]_2_, >95%) from Kwangduk Industry Co. served as the catalyst for gelation. Deionized water (DW) was utilized during the dilution of ethanol and sodium silicate.

TINUVIN 770 from BASF in the United States served as an amine light stabilizer, UV-326 from Rianlon Corporation in China functioned as a UV stabilizer, LOXIOL 8314 from Emery Oleochemicals in the Cincinnati, OH, USA was employed as both a lubricant and antistatic agent, and Zn-st from Duksan General Science in Korea was utilized as a lubricant.

### 3.2. Vinyl-Functionalized Mesoporous Silica Nanoparticles (V-MS)

In the first beaker, a silica precursor solution was prepared by mixing 600 mL of distilled water with sodium silicate solution. In the second beaker, a solution consisting of equal volumes of distilled water and ethanol (100 mL in total) was prepared, and its pH was adjusted to 3 using HCl. Subsequently, VTMS was added dropwise to the second beaker, and the hydrolysis reaction was induced over a period of 3 h. Upon the completion of the hydrolysis reaction, the hydrolyzed VTMS solution in the second beaker, which had a pH above 12 due to the presence of sodium silicate, was added dropwise to the first beaker where the sodium silicate solution had been diluted. The pH was adjusted to 10 using additional hydrochloric acid, and then, 20 mL of CTACl was added. The mixture was stirred at room temperature for 24 h. After the stirring was completed, the solution was centrifuged to obtain a residue as a white slurry. To remove CTACl, this slurry was dispersed in a 1.0 M solution of HCl/ethanol (EtOH) (1 L) and stirred for 24 h. After 24 h, another centrifugation was performed to collect a white slurry. This slurry was washed twice by dispersing it in a solution consisting of H_2_O/EtOH in a 1:1 volume ratio. Finally, the obtained slurry was dried in a convection oven at 80° for 10 h to yield white powder.

### 3.3. Manufacturing Method of Polyethylene Compound

Polyethylene compounds were manufactured using a Twin Extruder (TEK20-L/D48(9B)-2SF-1V-AC2.2, SM Platek Co., Ltd., Ansan-si, Republic of Korea). Recycled polyethylene (Kwangduk Industry Co., KD-rePE_030A) was fed into the main feeder, along with silica with incorporated vinyl functional groups and carbon black N330 (OCI, Seoul, Republic of Korea). TINUVIN 770 (BASF, Florham Park, NJ, USA) was utilized as an amine light stabilizer, UV-326 (Rianlon Corporation, Tianjin, China) as a UV stabilizer, LOXIOL 8314 (Emery Oleochemicals, Cincinnati, OH, USA) as both a lubricant and antistatic agent, and Zn-st (Duksan General Science, Seoul, Republic of Korea) as a lubricant. All of these were introduced through the side feeder as additives. During this process, the internal temperature of the extruder was maintained at 150–160 °C. Subsequently, the material was cooled and formed into pellet shapes. Afterward, the cross-linking reaction was carried out at 90 °C for 30 min using dicumyl peroxide ([C_6_H_5_C(CH_3_)_2_]_2_O_2_, Kwangduk Industry Co., Seoul, Republic of Korea), resulting in the final product of recycled PE with incorporated silica. When dicumyl peroxide is added in powder form and the temperature is raised, it undergoes a transition to a liquid state. To facilitate effective mixing, the powder of dicumyl peroxide and PE pellets were blended, and by utilizing a screw mechanism for agitation while increasing the temperature, the liquid dicumyl peroxide infiltrated the PE pellets, ensuring thorough blending. For this purpose, we conducted the cross-linking reaction at 90 °C.

### 3.4. Characterization of Vinyl-Functionalized Mesoporous Silica Nanoparticles

The morphology and particle shape of the synthesized mesoporous silica were examined using a Field Emission Transmission Electron Microscope (FE-TEM; HF-3300) under 300 kV conditions. Additionally, the specific surface area and porosity characteristics of the porous silica were analyzed using a nitrogen adsorption–desorption isotherm (N_2_-sorption; QUANTACHROME, Qudrasorb SI). The measurements were conducted at a temperature of 77 K, maintained using liquid nitrogen. The adsorbed nitrogen was normalized to standard temperature and pressure. Prior to analysis, a heat treatment was performed at 200 °C for 6 h to remove moisture and impurities adsorbed on the sample’s surface. The Brunauer–Emmett–Teller (BET) specific surface area was calculated from the linear portion (P/P_0_ = 0.05–0.30) of the BET equation. The volume and size of pores were calculated using the Barrett–Joyner–Halenda (BJH) equation. For confirmation of the properties of the mesopores, an X-ray diffractometer (XRD; PANalytical X’pert PRO MRD) was employed. Measurements were conducted in 2θ scan mode with Cu-Kα rays (λ = 0.0154 nm). Finally, to confirm the introduction of vinyl functional groups, Fourier Transform Infrared Spectrometry (FTIR; Nicolet iS50, Thermo, Waltham, MA, USA) was used for analysis.

### 3.5. Characterization of Polyethylene Compounds

To evaluate the physical performance of recycled PE, analyses were conducted using the Melting Index test (QM280, Qmesys, Uiwang-si, Republic of Korea) and a Universal Testing Machine (UTM, QM100s, Qmesys, Republic of Korea). The Melting Index was carried out in accordance with ASTM D123 standards (Standard Terminology Relating to Textiles). Using the UTM, various tests were conducted, including tensile strength, elongation, elastic modulus, flexural strength, and flexural modulus. Tensile strength, elongation, and elastic modulus were assessed by preparing specimens according to ASTM D638 standards (Standard Test Method for Tensile Properties of Plastics). The tests were performed with a testing speed of 50 mm/min, a grip-to-grip distance of 115 mm, a gauge length of 50 mm, and a load cell capacity of 30,000 N. Flexural strength and flexural modulus were evaluated based on ASTM D790 standards. Specimens were prepared, and the tests were conducted with a testing speed of 2.8 mm/min, a grip-to-grip distance of 104 mm, and an elastic range of 0.05% to 0.25%.

## 4. Conclusions

In this study, we synthesized vinyl-functionalized porous silica using sodium silicate and vinyltrimethoxysilane without the need for polymerization or sintering processes. Unlike conventional approaches that introduce vinyl groups after synthesizing porous silica, our method allowed for the synthesis of high-specific-surface-area silica with incorporated vinyl functionalities through a simplified process. This suggests the potential for commercialization beyond our current research.

The synthesized vinyl-functionalized silica was incorporated into the production of recycled PE, and the physical characteristics were compared. The results revealed that higher specific surface areas and increased vinyl content correlated with improved physical properties. Unlike the typical introduction of nano-silica into recycled PE, where only tensile strength increased at the expense of flexibility, the incorporation of vinyl functional groups induced chemical bonding within the inorganic silica. This resulted in an increase in cross-linking density, enhancing not only tensile strength but also other flexibility-related performance indicators.

Our findings confirm the potential of vinyl-functionalized porous silica and inspire further research into its application in various recycled plastics and rubber. This study concludes by highlighting the promising performance enhancements achievable through the introduction of vinyl-functionalized porous silica.

## Figures and Tables

**Figure 1 ijms-25-02295-f001:**
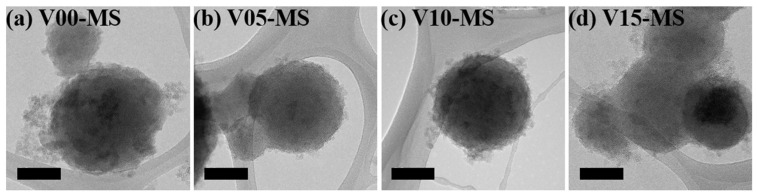
TEM images of V-MS nanoparticles (All scale bars are 100 nm).

**Figure 2 ijms-25-02295-f002:**
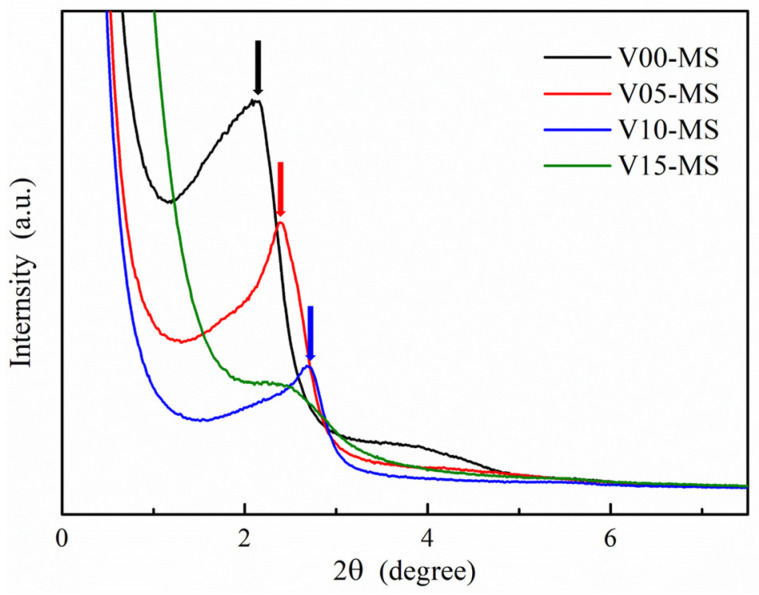
Low-angle XRD analysis of vinyl-functionalized mesoporous silica nanoparticles.

**Figure 3 ijms-25-02295-f003:**
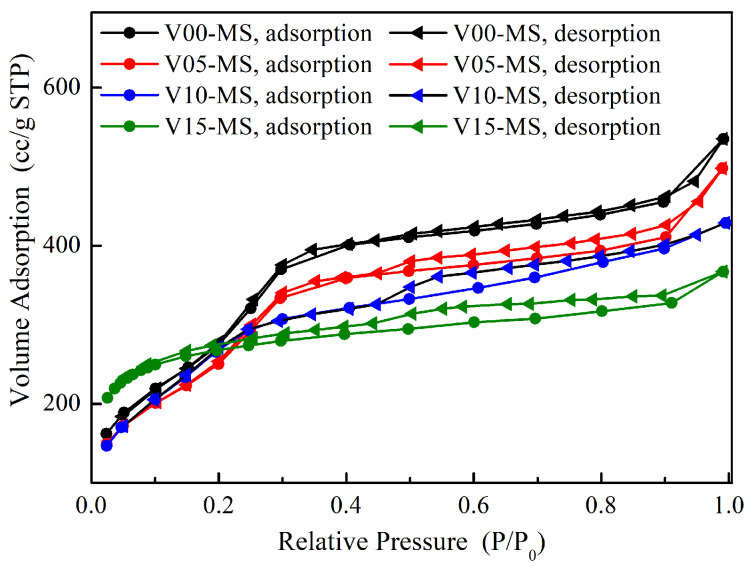
N_2_ adsorption–desorption isotherm graphs of V-MS nanoparticles.

**Figure 4 ijms-25-02295-f004:**
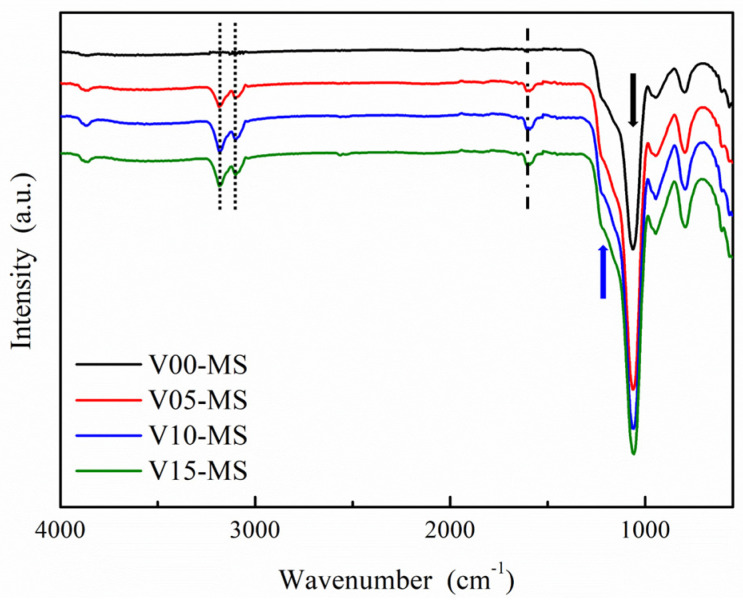
FTIR graphs of V-MS.

**Figure 5 ijms-25-02295-f005:**
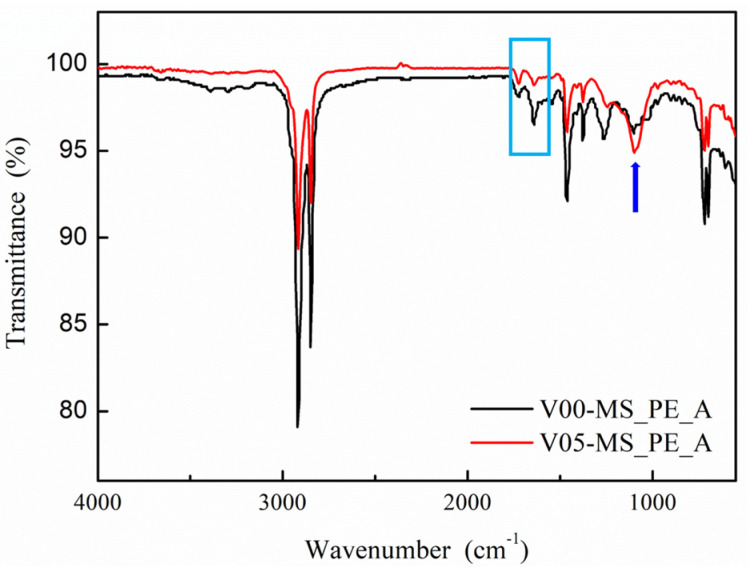
FTIR graphs of recycled polyethylene with mesoporous silica.

**Table 1 ijms-25-02295-t001:** Volume ratio of sodium silicate to VTMS in the synthesis of vinyl-functionalized mesoporous silica (V-MS).

	Sodium Silicate (mL)	VTMS (mL)
V00-MS	40	0
V05-MS	35	5
V10-MS	30	10
V15-MS	25	15

**Table 2 ijms-25-02295-t002:** The textural properties of V-MS nanoparticles.

	BET (m^2^/g)	Pore Size (nm)	Pore Volume (cc/g)
V00-MS	1263	3.826	0.387
V05-MS	1139	3.486	0.463
V10-MS	1012	3.168	0.513
V15-MS	883	3.817	0.626

**Table 3 ijms-25-02295-t003:** Characterization of recycled polyethylene compounds with various V05-MS NP contents.

Sample	Melt Index (g/10 min)	Tensile Strength (kg/cm^2^)	Elongation (%)	Elastic Modulus (kg/mm^2^)	Flexural Strength (kg/cm^2^)	Flexural Modulus (kg/mm^2^)
STD PE_A rePE_A 100phr V05-MS 0 phr *	2.6 ±0.5	97.1 ±9.7	113.6 ±11.3	15.3 ±1.5	48.5 ±4.8	14.0 ±1.4
V05-MS_PE_A #1 rePE_A 100phr V05-MS 1 phr	2.3 ±0.5	100.5 ±10.0	101.8 ±10.1	15.7 ±1.5	50.1 ±5.0	13.9 ±1.3
V05-MS_PE_A #2 rePE_A 100 phr V05-MS 2 phr	1.3 ±0.5	113.1 ±11.3	84.1 ±8.4	9.7 ±0.9	65.8 ±6.5	7.2 ±0.7
V05-MS_PE_A #3 rePE_A 100 phr V05-MS 3 phr	0.6 ±0.5	124.9 ±12.4	70.2 ±7.0	5.4 ±0.5	77.4 ±7.7	3.3 ±0.3

* “phr” means Part per Hundred Rubber.

**Table 4 ijms-25-02295-t004:** Characterization of recycled polyethylene compounds according to the content of the vinyl functional group introduced into mesoporous silica.

Sample	Melt Index (g/10 min)	Tensile Strength (kg/cm^2^)	Elongation (%)	Elastic Modulus (kg/mm^2^)	Flexural Strength (kg/cm^2^)	Flexural Modulus (kg/mm^2^)
**STD PE_A** rePE_A 100phr V05-MS 0 phr *	2.6 ±0.5	97.1 ±9.7	113.6 ±11.3	15.3 ±1.5	48.5 ±4.8	14.0 ±1.4
**V00-MS_PE_A** rePE_A 100phr V00-MS 1 phr	2.5 ±0.5	101.1 ±8.6	90.6 ±8.0	14.4 ±0.6	41.8 ±4.1	10.6 ±0.8
**V05-MS_PE_A** rePE_A 100phr V05-MS 1 phr	2.3 ±0.5	100.5 ±10.0	101.8 ±10.1	15.7 ±1.5	50.1 ±5.0	13.9 ±1.3
**V10-MS_PE_A** rePE_A 100phr V10-MS 1 phr	2.4 ±0.5	109.5 ±10.2	121.4 ±12.1	20.5 ±2.0	65.6 ±6.5	16.0 ±1.6
**V15-MS_PE_A** rePE_A 100phr V15-MS 1 phr	2.3 ±0.5	102.1 ±10.2	103.3 ±10.3	17.7 ±1.7	60.2 ±6.0	11.0 ±1.1

* “phr” means Part per Hundred Rubber.

## Data Availability

Data is contained within the articl.

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
