# Peer review of "The Facile Synthesis and Application of Mesoporous Silica Nanoparticles with a Vinyl Functional Group for Plastic Recycling"

_ijms, 2024, doi:10.3390/ijms25042295_

Round 1

Reviewer 1 Report

Comments and Suggestions for Authors

In the manuscript submitted for review, the authors present a study on synthesizing silica nanoparticles with vinyl functional groups and evaluating their mechanical performance when used in recycled plastics.

 General comments

 ·       The manuscript contains descriptions of studies that have been well planned, carried out and interpreted.

·       The manuscript is quite well related to existing literature but can be improved.

·       There are sufficient details given to replicate the proposed experimental procedures for synthesis of the

·       The purpose of the study is stated clearly.

·       Conclusions are stated clearly.

The results are interesting to readers, but some questions/recomandations should be addressed before I recommend it for publication.

The specific comments are as follows:

The interpretation of the XRD results - it is not indicated the used JCPDS card.

In the Figure 2, Figure 4, Figure 5: The peaks are not indexed.

No comparison is made with similar results existing in other works.

section 3.1 Materials

- is not indicated in the concentration of the reactants.

section 3.2 Vinyl functionalized mesoporous silica nanoparticles (V-MS):

-       the concentration of the reactants used is not specified: sodium silicate solution, hydrochloric acid, Cetyltrimethylammonium chloride.

 section 3.4 Characterization of Vinyl functionalized mesoporous silica nanoparticles:

- a more detailed description of the conditions for the characterization of the obtained materials is recommended.

Reviewer 2 Report

Comments and Suggestions for Authors

Dear authors, 
This is a good piece of research. However, the synthesis of V-MS is not at all clear as  the amount of sodium silicate used is confusing. For synthesis it is used as a solution, but have you prepared it or is it commercial? What is the concentration of the solution?

Reviewer 3 Report

Comments and Suggestions for Authors

Dear Authors

I have reviewed your paper entitled "The facile synthesis and application of mesoporous silica nanoparticles with a vinyl functional group for plastic recycling" Which investigated the effect of mesoporous silica with vinyl group adding on the mechanical performance of recycled polyethylene, the experimental data is abundant and the interpretation of the results is generally accurate.

I think the following parts should be explained.

1. Introduction:

The author should explain why VTMS is used. what are the advantages by applying VTMS rather than tetraethylorthosilicate?

2. line 113-115.

“The vinyl silane only possesses three alkoxysilane groups, leading to uneven silane polymerization when compared to other samples”. Please explain why.

3. Mesoporous silica possesses high specific surface area, and the vinyl groups were introduced into porous silica by using VTMS. Theoretically, vinyl group modified silica can also be prepared with VTMS as precursor in this reported experimental process. Thus, whether vinyl functionalized mesoporous silica is a mixture of mesoporous silica and vinyl group modified silica.

4. How to calculate the content of the vinyl functional group introduced into mesoporous silica.

5. V10-MS 1phr exhibits best mechanical performance, characterization of recycled polyethylene compounds with various V10-MS contents should be conducted.

Round 2

Reviewer 3 Report

Comments and Suggestions for Authors

The manuscript is now acceptable for publication.